# Induction of BVR-A Expression by Korean Red Ginseng in Murine Hippocampal Astrocytes: Role of Bilirubin in Mitochondrial Function via the LKB1–SIRT1–ERRα Axis

**DOI:** 10.3390/antiox11091742

**Published:** 2022-09-01

**Authors:** Sunhong Moon, Chang-Hee Kim, Jinhong Park, Minsu Kim, Hui Su Jeon, Young-Myeong Kim, Yoon Kyung Choi

**Affiliations:** 1Bio/Molecular Informatics Center, Department of Bioscience and Biotechnology, Konkuk University, Seoul 05029, Korea; 2Department of Otorhinolaryngology-Head and Neck Surgery, Konkuk University Medical Center, Konkuk University School of Medicine, Seoul 05029, Korea; 3Department of Molecular and Cellular Biochemistry, School of Medicine, Kangwon National University, Chuncheon 24341, Korea

**Keywords:** liver kinase B1, biliverdin reductase, antioxidants, bilirubin, mitochondrial membrane potential, oxidative phosphorylation, superoxide dismutase 2

## Abstract

The beneficial effects of Korean red ginseng extract (KRGE) on the central nervous system (CNS) have been reported. Among the CNS cells, astrocytes possess robust antioxidative properties and regenerative potential. Under physiological conditions, biliverdin reductase A (BVR-A) converts biliverdin (a heme oxygenase metabolite) into bilirubin, a major natural and potent antioxidant. We found that KRGE enhanced BVR-A in astrocytes in the fimbria region of the adult mouse hippocampus under physiological conditions. KRGE-induced BVR-A expression and subsequent bilirubin production were required for changes in mitochondrial membrane potential, mitochondrial mass, and oxidative phosphorylation through liver kinase B1 (LKB1), estrogen-related receptor α (ERRα), and sirtuin (SIRT1 and SIRT5) in astrocytes. However, BVR-A did not affect the KRGE-induced expression of AMP-activated protein kinase α (AMPKα). The KRGE-stimulated BVR-A–LKB1–SIRT1–ERRα pathway regulates the levels of mitochondria-localized proteins such as SIRT5, translocase of the outer mitochondrial membrane 20 (Tom20), Tom22, cytochrome c (Cyt c), and superoxide dismutase 2 (SOD2). Increased Tom20 expression in astrocytes of the hippocampal fimbria region was observed in KRGE-treated mice. KRGE-induced expression of Cyt c and SOD2 was associated with the Tom20/Tom22 complex. Taken together, KRGE-induced bilirubin production is required for enhanced astrocytic mitochondrial function in an LKB1-dependent and AMPKα-independent manner under physiological conditions.

## 1. Introduction

Under physiological conditions, Korean red ginseng extract (KRGE) promotes astrocyte proliferation in the brain of young adult mice [1]. Enhanced mitochondrial function is associated with cell proliferation [2], possibly through the heme oxygenase (HO)-mediated pathway [1]. Reactive oxygen species can be produced in the process of oxidative phosphorylation (OXPHOS) in mitochondria. Superoxide dismutase 2 (SOD2) protects mitochondria from undesirable oxidative effects [3]. Biliverdin reductase A (BVR-A) is an enzyme that converts biliverdin into bilirubin [4]; however, its specific functions in adults have not been clarified. The physiological antioxidant effects of bilirubin depend on the expression and activity of BVR-A [5].

The serine/threonine kinase liver kinase B1 (LKB1), a principal regulator of energy metabolism, inhibits cancer cell growth under energy stress conditions through the modulation of 14 kinases that act on various processes, such as cell attachment and metabolism [6]. One of the target proteins regulated by LKB1 is the AMP-activated protein kinase α (AMPKα) [7], which is important for mitochondrial biogenesis activated by HO metabolites (e.g., carbon monoxide (CO)) [8,9]. After CO pretreatment, followed by a recovery phase (CO/R), AMPKα can activate the peroxisome-proliferator-activating receptor-γ coactivator-1α (PGC-1α)-estrogen-related receptor α (ERRα) pathway in astrocytes [8].

In the outer mitochondrial membrane, the translocase of the outer mitochondrial membrane (TOM) complex leads to the primary entry of nuclear-encoded proteins into mitochondria [10]. Tom20 and Tom22, together with other TOM complexes, bind to the outer membrane of mitochondria [6] and play key roles in the regulation of the mitochondrial membrane potential and protein import [11]. Mitochondrial quality control stems from sirtuins (SIRTs). In mammals, seven SIRTs are classified as SIRT1–7; specifically, SIRT1 is localized in the cytosol and the nucleus, SIRT2 and SIRT4 are localized in the cytosol and mitochondria, SIRT3 and SIRT5 are localized in the mitochondria, and SIRT6 and SIRT7 are localized in the nucleus [12,13,14,15]. SIRTs are nicotinamide adenine dinucleotide (NAD^+^)-dependent deacetylases [15].

Here, we investigate the roles of KRGE in mitochondrial functions (i.e., mitochondrial membrane potential, mitochondrial mass, and expression of OXPHOS, Tom20, Tom22, cytochrome c (Cyt c), and SOD2) under physiological conditions. Our novel findings show that KRGE induces BVR-A expression in astrocytes in vitro and hippocampal astrocytes, especially in the fimbria region, in vivo. BVR-A-derived bilirubin is associated with KRGE-induced mitochondrial function changes via the LKB1–SIRT1–ERRα pathway. The interplay between cytosolic localization of SIRT1 and mitochondrial localization of SIRT5 may facilitate mitochondrial functions by regulating Tom20 and Tom22, consequently upregulating Cyt c and SOD2. In addition, the combined functions of Tom20 and Tom22 facilitate SIRT5 expression. Taken together, these results indicate that KRGE-induced BVR-A/bilirubin is critical for astrocytic mitochondrial function.

## 2. Materials and Methods

### 2.1. Materials

We obtained mannitol, sucrose, MOPS, and ethylene-diamine-tetra acetic acid (EDTA) from Sigma-Aldrich (St. Louis, MO, USA). Fetal bovine serum (FBS) was obtained from HyClone Laboratories (Omaha, NE, USA). The proteinase inhibitor cocktail was purchased from Thermo Fisher Scientific (Waltham, MA, USA). The SIRT activity inhibitor nicotinamide (NAM) was purchased from Merck Millipore (Darmstadt, Germany). Bilirubin, biliverdin, CORM-2 ([Ru(CO)_3_Cl_2_]_2_, a CO-releasing compound), and FeCl_2_ were purchased from Sigma-Aldrich. Bilirubin powder was dissolved in 0.1 M NaOH to make a 50 mM stock solution and stored at −25 °C in the dark. CORM-2 (100 mM stock) and biliverdin (50 mM stock) were dissolved in DMSO (Sigma-Aldrich) and stored at −75 °C. KRGE was obtained from the Korea Ginseng Corporation and stored at 4 °C. Stock solutions were prepared by filtering distilled water. The aliquoted stock solutions (250 mg/mL) were stored at −25 °C in the dark.

### 2.2. Cell Culture

Primary human brain astrocytes were acquired from Applied Cell Biology Research Institute (Kirkland, WA, USA). Astrocytes were cultured in Dulbecco’s modified Eagle’s medium (DMEM, HyClone, Omaha, NE, USA) supplemented with 10% FBS (Corning, NY, USA) and 1% penicillin/streptomycin (HyClone).

### 2.3. Treatment of Cells with Reagents

Astrocytes were cultured to reach a density of 80%, and the medium was changed to 0% FBS (serum-free) DMEM for 3 h. Then, 50 µM bilirubin, 50 µM biliverdin, 50 µM CORM-2, or 50 µM FeCl_2_ was added for 4 h.

### 2.4. Transfection

At 70% confluency, astrocytes were transfected with small interfering RNAs (siRNAs) against SIRT1, SIRT5, ERRα, HO-1, LKB1, Ca^2^^+^/calmodulin-dependent protein kinase kinase α (CaMKKβ), Tom20, Tom22, BVR-A (50 nM, Santa Cruz Biotechnology, Dallas, TX, USA), or a negative control (50 nM; Thermo Fisher Scientific) using RNAiMax (Thermo Fisher Scientific). After approximately 16 h of recovery, cells were incubated with or without KRGE for 24 h in serum-free DMEM.

### 2.5. MitoTracker Staining

Intracellular active mitochondrial levels were measured by quantitative fluorescence imaging using the mitochondrial membrane potential-sensitive dye, MitoTracker^®^ Deep Red FM (Thermo Fisher Scientific). Astrocytes plated on 25 mm round coverslips in 6-well plates were cultured until they reached 80% confluence. The medium was changed to serum-free DMEM for 3 h. Cells were then treated with 50 µM HO metabolites (biliverdin, CORM-2 (CO), or FeCl_2_ (Fe^2+^)) and 50 µM bilirubin for 4 h. After 3 h of starvation, astrocytes were incubated with 5 mM NAM for 30 min and 50 µM bilirubin for 3.5 h, followed by 0.5 µM MitoTracker-Red (Thermo Fisher Scientific) for 0.5 h. Phosphate-buffered saline (PBS) was used for washing the cells. Fluorescent images of live cells were obtained using a confocal microscope (Olympus FV1000), and the intensity was obtained and analyzed using ImageJ (https://rsb.info.nih.gov/ij/ accessed on 1 March 2022). The average intensity of five randomly chosen cells from each image was determined.

### 2.6. Mitochondria-Enriched Fraction

Cells in four 100 mm dishes were used for each group. Water or 0.5 mg/mL KRGE was added for 24 h, and the cells were washed with PBS and collected using mannitol–sucrose buffer containing 210 mM mannitol, 70 mM sucrose, 5 mM MOPS, 1 mM EDTA, and a protease inhibitor cocktail at pH 7.4. After the collected cells were homogenized (with a 27-gauge 1/2-inch needle, the homogenates were centrifuged at 800× *g* for 10 min at 4 °C to pellet the nuclei. The supernatants were used as total lysates or centrifuged at 10,000× *g* for 20 min at 4 °C to collect mitochondria-enriched fraction pellets. The pellets were washed with mannitol–sucrose buffer and centrifuged at 10,000× *g* for 20 min at 4 °C, and the supernatants were filtered with a 3 kDa centrifugal filter (Merck Millipore, Darmstadt, Germany) at 4000× *g* for 2 h (cytosolic protein). The mitochondria-enriched fraction (pellet) was suspended in 30 µL Protein Extraction Solution (RIPA buffer; Elpis-Biotech, Daejeon, South Korea), incubated for 20 min on ice, and centrifuged at 15,000× *g* for 20 min at 4 °C. Mitochondrial protein (10 µg, supernatant solution) and cytosolic protein (10 µg) were subjected to Western blotting after quantification using Pierce BCA protein assay reagent (Thermo Fisher Scientific). Antibodies directed against the following proteins were used: SIRT1, SIRT5, Tom20, Tom22, SOD2, voltage-dependent anion channel 1 (Santa Cruz Biotechnology), β-actin (Sigma-Aldrich), and Cyt c (BD Pharmingen).

### 2.7. Immunocytochemistry

For ERRα immunocytochemistry, human astrocytes were seeded on coverslips in 12-well (Falcon) incubation plates. Astrocytes were fixed in 4% formaldehyde for 20 min at 23 ± 2 °C, washed gently with 0.1% (*v*/*v*) Tween 20 in PBS for 10 min each, blocked with 3% bovine serum albumin (BSA) (USBiological Life Sciences, Salem, MA, USA), diluted in 0.1% Triton X-100 (Sigma-Aldrich) in PBS for 1 h at room temperature, and incubated with the ERRα primary antibody (1:150, Santa Cruz Biotechnology) overnight at 4 °C, followed by incubation with an Alexa Fluor antibody (1:600, Thermo Fisher Scientific). Nuclei were stained with 4′,6-diamidino-2-phenylindole (DAPI) (Thermo Fisher Scientific). Images were obtained using a confocal microscope (Olympus FV1000; Tokyo, Japan). For SOD2 and Tom20 immunoreactivity, samples were incubated with rabbit anti-SOD2 (1:300, Santa Cruz Biotechnology) and mouse anti-Tom20 (1:500, Abcam, Cambridge, UK) overnight at 4 °C. After washing, the secondary antibody was diluted in 0.3% BSA and added to each well. The samples were incubated with a mixture of tetramethylrhodamine (TRITC)-conjugated donkey immunoglobulin G (IgG) (1:300; Jackson ImmunoResearch, West Grove, PA, USA) and fluorescein isothiocyanate (FITC)-conjugated donkey IgG (1:300; Jackson ImmunoResearch, West Grove, PA, USA) for 1 h at room temperature. After the final wash, the cells were mounted on microslides using Fluoro-Gel II with DAPI mounting solution (Electron Microscopy Sciences, Hatfield, PA, USA). The upper side of the coverslips was reversed to allow for attachment to the DAPI mounting solution. Images were acquired using an inverted microscope (Eclipse Ti2-U; Nikon, Tokyo, Japan).

### 2.8. Detection of Oxidative Phosphorylation

Protein (10–15 µg) from the cell lysates (lysed by RIPA) was combined with sodium dodecyl sulfate (SDS) sample buffer (10% glycerol (*v*/*v*), Tris-Cl pH 6.8, 2% SDS (*w*/*v*), 1% β-mercaptoethanol (*v*/*v*), and bromophenol blue) and heated at 37 °C for 5 min. Protein samples were then subjected to Western blot analysis. OXPHOS antibody (1:8000; Abcam, Cambridge, UK) was used. The OXPHOS antibody detects the oxidative phosphorylation complex I (NDUFB8, NADH: ubiquinone oxidoreductase subunit B8), complex II (succinate dehydrogenase complex iron-sulfur subunit B), complex III (UQCRC2, ubiquinol-cytochrome c reductase core protein 2), complex IV (MTCO1, cytochrome c oxidase subunit I), and complex V (ATP5A, ATP synthase).

### 2.9. Western Blot Analysis

RIPA buffer was used for cell lysis. Brain tissue samples were obtained from bregma −1.1 to −1.3, and cells were lysed in RIPA buffer. Selected amounts of protein from the cell lysates were combined with SDS sample buffer (10% glycerol (*v*/*v*), Tris-Cl pH 6.8, 2% SDS (*w*/*v*), 1% β-mercaptoethanol (*v*/*v*), and bromophenol blue) and incubated at 100 °C for 5 min. Protein samples were then separated using SDS polyacrylamide gel electrophoresis, transferred to polyvinylidene difluoride membranes (MilliporeSigma, Burlington, MA, USA), and blocked using Tris-buffered saline containing 0.1% Tween 20 and 5% skim milk (BD Difco, BD, Franklin Lakes, NJ, USA). The membranes were incubated with primary antibodies at 4 °C overnight. The primary antibodies used in this study were as follows: Tom20 (1:1000, Santa Cruz Biotechnology, Dallas, TX, USA; 1:5000, Abcam), Tom22 (1:1000, Santa Cruz Biotechnology), Cyt c (1:3000, BD Pharmingen, San Jose, CA, USA), SIRT1 (1:1000, Santa Cruz Biotechnology), SIRT2 (1:5000, Abcam), SIRT3 (1:1000, Santa Cruz Biotechnology), SIRT4 (1:3000, Thermo Fisher Scientific), SIRT5 (1:1000, Santa Cruz Biotechnology), ERRα (1:3000, Novus Biologicals, Littleton, CO, USA), p-LKB1 (Ser^428^) (1:2000, Cell Signaling Technology, Danvers, MA, USA), LKB1 (1:2000, Cell Signaling Technology), CaMKKβ (1:1000, Santa Cruz Biotechnology), AMPKα (1:2000, Cell Signaling Technology), p-AMPKα (Thr^172^) (1:2000, Cell Signaling Technology), SOD2 (1:1000, Santa Cruz Biotechnology), mitochondria (1:10,000, Abcam), HO-1 (1:2000, BD Biosciences, Franklin Lakes, NJ, USA), BVR-A (1:1000, Santa Cruz Biotechnology; 1:3000, Enzo Life Sciences, Farmingdale, NY, USA), and anti-β-actin (1:8000, Sigma-Aldrich). After washing, the membranes were incubated with peroxidase-conjugated secondary antibodies (Thermo Fisher Scientific) and visualized using enhanced chemiluminescence (Elpis-Biotech) with the appropriate Western blot detection equipment (Fusion Solo, Vilber, Collégien, France).

### 2.10. Cellular Reactive Oxygen Species (ROS) Detection

For ROS detection, human astrocytes were seeded into 12-well plates (Falcon, Glendale, CA, USA) and transfected with specific siRNAs. After approximately 16 h, the cells were treated with water or 0.5 mg/mL KRGE for 23.5 h in serum-free DMEM media. Then, the astrocytes were treated with 20 μM H_2_DCFDA (Thermo Fisher Scientific) for 30 min at 23 ± 2 °C. Finally, the cells were washed gently with PBS for 10 min. Images were captured using an inverted microscope (Eclipse Ti2-U; Nikon).

### 2.11. Intracellular ATP Levels

An ATP colorimetric assay kit (Bio-Vision, Milpitas, CA, USA) was used for measuring intracellular ATP levels. Human astrocytes cultured on a 60 mm dish (Falcon) were transfected with indicated siRNAs and exposed to water or 0.5 mg/mL KRGE for 24 h in serum-free DMEM media. After trypsinization, the cells were transferred to Eppendorf tubes. The cell pellet was lysed in 120 μL of ATP assay buffer and incubated for 5 min at room temperature. The tubes were centrifuged at 15,000 rpm for 5 min at 4 °C. Next, the supernatant (50 μL) was mixed with reaction mixture reagents (50 μL) on 96-well plates (Falcon). After a 30 min incubation with blocking light at room temperature, absorbance was measured at 570 nm (Epoch Microplate Spectrophotometer, BioTek, Santa Clara, CA, USA). Lysed cells were quantified with BCA reagents (Thermo Fisher Scientific, Carthage, MO, USA). ATP levels and protein amount (nmol ATP/mg protein) in the control group were set as “1”, and the other groups were adjusted to the control group.

### 2.12. Animals and Brain Tissue Immunohistochemistry

Male C57BL/6 mice were purchased from Joong Ah Bio Inc. (Suwon, South Korea) and maintained under standard conditions, with water and food available ad libitum. KRGE (0.05, 0.125, or 0.25 mg/mL) was administered via the drinking water for 3 days. The control group was administered only water. For histological analysis, mice were anesthetized using isoflurane (1.5%) and N_2_O gas and then transcardially perfused with saline. Using an optimal cutting temperature compound, frozen brain tissues were sliced into 20 μm sections on a cryostat (HM525 NX, Thermo Fisher Scientific). The sections were incubated with 4% paraformaldehyde for 15 min, washed three times in PBS, and then incubated with 3% BSA for 1 h. The sections were then incubated with rabbit anti-Tom20 (1:300, Abcam), mouse anti-glial fibrillary acidic protein (GFAP; 1:300, BD Bioscience), and anti-BVR-A (1:200, Enzo Life Sciences) antibodies in 0.1% Triton X-100 in PBS at 4 °C overnight. After washing, the sections were incubated in a mixture of TRITC-conjugated donkey anti-rabbit IgG (1:300, Jackson ImmunoResearch) and FITC-conjugated donkey anti-mouse IgG (1:300, Jackson ImmunoResearch) for 1 h at room temperature. Between incubations, tissues were washed with 0.1% Tween 20 in PBS. The sections were visualized using a mounting solution (Fluoro-Gel II with DAPI; Electron Microscopy Sciences). The stained sections were examined under an inverted phase-contrast microscope (Eclipse Ti2eU, Nikon).

### 2.13. Data Analysis

Quantification of the intensity of protein expression in Western blotting and immunocytochemistry was performed using ImageJ. GraphPad Prism 6 was used for statistical analyses. Multiple comparisons were evaluated using one-way analysis of variance (ANOVA) and Tukey’s post hoc test (mean ± standard deviation).

## 3. Results

### 3.1. BVR-A and GFAP Are Co-Expressed in the Fimbria Region of the Mouse Hippocampus by KRGE

Compared with water-administered mice brains, KRGE-administered mice brains showed higher co-localization of BVR-A immunoreactivity with the astrocyte marker GFAP in the fimbria region of the hippocampus (approximately bregma −1.7), demonstrating that GFAP-stained astrocytes express BVR-A (Figure 1a,b). Protein levels of BVR-A were increased by KRGE in a dose-dependent manner in mouse brains (Figure 1c) and human astrocytes (Figure 1d). Next, we investigated the relationship between BVR-A and HO-1 in astrocytes because BVR-A may act as a transcription factor for HO-1 during oxidative stress [16]. HO-1 is associated with astrocytic energy production through AMPKα-ERRα-mediated mitochondrial functions [11,17]. Specific knockdown of BVR-A in KRGE-treated astrocytes did not significantly change HO-1 expression (Figure 1e) and vice versa (Figure 1f), implying that BVR-A may act as a bilirubin-producing enzyme without affecting HO-1 expression in physiological brains.

### 3.2. Among HO Metabolites, Bilirubin Significantly Induces ERRα Expression and Mitochondrial Function

Since KRGE upregulates HO-1 and BVR-A protein levels (Figure 1e,f), we investigated the role of HO metabolites (i.e., biliverdin, CO, and Fe^2+^) and bilirubin on mitochondrial function. Among them, bilirubin markedly increased the mitochondrial membrane potential detected by MitoTracker-Red (Figure 2a) and ERRα localization in the nucleus (Figure 2b). These results suggest that the conversion of biliverdin to bilirubin may be a key event in mitochondrial function. Therefore, we compared the effects of direct bilirubin and biliverdin on astrocytic mitochondrial function. A comparison between bilirubin and biliverdin was performed by detecting mitochondria-related proteins such as ERRα, mitochondria (Mito), Cyt c, SOD2, and OXPHOS in human astrocytes (Figure 2c). A significant increase in ERRα, Mito, Cyt c, and SOD2 was detected following the administration of bilirubin but not biliverdin (Figure 2c and Appendix A). The increase in mitochondrial functional proteins was blocked by ERRα knockdown; however, BVR-A expression was not changed by diminished ERRα levels in the presence of KRGE (Figure 2d). These results suggest that the KRGE-stimulated BVR-A–bilirubin–ERRα axis upregulates proteins related to mitochondrial mass (i.e., Mito), OXPHOS (i.e., Cyt c), and antioxidative stress (i.e., SOD2) under physiological conditions.

### 3.3. KRGE Induces BVR-A/Bilirubin-Mediated LKB1 Activation

To confirm the effect of BVR-A-mediated bilirubin production on ERRα expression induced by KRGE under physiological conditions, astrocytes were transfected with BVR-A siRNA. Diminished BVR-A levels resulted in a significant reduction in KRGE-induced ERRα expression, consequently leading to the downregulation of Mito, Cyt c, and SOD2 (Figure 3a). We next investigated the mechanisms linking bilirubin production and ERRα upregulation in KRGE-treated astrocytes. A recent report demonstrated that LKB1-deficient astrocytes exhibit impaired metabolic function and enhanced inflammatory activation [18]. Interestingly, bilirubin increased LKB1 phosphorylation in a concentration-dependent manner (Figure 3b). Biliverdin increased BVR-A levels (Figure 3b), suggesting that it can be converted to bilirubin following BVR-A enzyme induction. Diminished BVR-A expression, in the presence of biliverdin, reduced p-LKB1 levels (Figure 3c), confirming that bilirubin is critical for LKB1 activation. In a concentration-dependent manner, KRGE induced p-LKB1 expression in both in vitro astrocytes (Figure 3d) and in vivo brain tissues (Figure 3e). Therefore, the KRGE-induced upregulation of bilirubin may result in LKB1 activation.

### 3.4. KRGE Induces the LKB1-Mediated Expression of ERRα and Mitochondrial Functional Proteins

To confirm the effect of KRGE-induced bilirubin production on LKB1 activation, human astrocytes were transfected with BVR-A siRNA under KRGE-treated normal conditions. Treatment of astrocytes with BVR-A siRNA markedly reduced KRGE-induced upregulation of p-LKB1 levels and ratios of p-LKB1 to LKB1 (Figure 4a). LKB1 phosphorylates and activates multiple downstream kinases (e.g., AMPKα) and regulates metabolism and mitochondrial biogenesis [19,20]. Therefore, we evaluated the expression of p-AMPKα and the p-AMPK/AMPKα ratio in KRGE-treated cells with decreased BVR-A levels. Knockdown of BVR-A did not significantly change KRGE-induced p-AMPKα levels and the p-AMPK/AMPKα ratio, even though KRGE increased both p-AMPKα levels and the p-AMPK/AMPKα ratio (Figure 4a). AMPKα is phosphorylated and activated by CaMKKβ and LKB1 [21]. Considering that KRGE-induced upregulation of p-AMPKα may stem from CaMKKβ and/or LKB1 activation, human astrocytes were transfected with specific siRNAs targeting CaMKKβ or LKB1. Knockdown of both LKB1 and CaMKKβ synergistically downregulated ERRα, p-AMPKα, Mito, Cyt c, and SOD2 levels (Figure 4b,c). LKB1 knockdown alone significantly reduced the KRGE-induced upregulation of ERRα (Figure 4b), Mito, Cyt c, and SOD2 (Figure 4c) levels. By contrast, CaMKKβ knockdown alone resulted in the reduction of KRGE-induced Mito and SOD2 levels (Figure 4c), suggesting that KRGE activates both LKB1 and CaMKKβ. Taken together, KRGE stimulation of the BVR-A/bilirubin pathway results in the LKB1–ERRα-axis-mediated upregulation of mitochondrial functional proteins (i.e., Mito, Cyt c, and SOD2). In addition to this pathway, KRGE can also induce CaMKKβ-mediated Mito and SOD2 expression.

### 3.5. Bilirubin Increases Mitochondrial Membrane Potential through SIRT Activation

Bilirubin-induced increases in mitochondrial membrane potential were significantly reduced by the SIRT inhibitor NAM (Figure 4d). Because SIRT1–5 proteins are related to mitochondrial functions [9,12,13,15], protein levels of SIRT1 through SIRT5 were evaluated in KRGE-treated cells (Figure 4e). KRGE administration significantly increased SIRT1 and SIRT5 protein levels (Figure 4e). We found that KRGE-induced LKB1 activation was related to SIRT1 upregulation without affecting SIRT2, SIRT3, and SIRT4 expression (Figure 4e), suggesting the involvement of the LKB1–SIRT1 axis. Knockdown of LKB1 partly reduced KRGE-induced SIRT5 expression; however, knockdown of CaMKKβ significantly reduced KRGE-induced SIRT5 expression without affecting SIRT1, SIRT2, SIRT3, and SIRT4 levels (Figure 4e). In addition, the expression levels of SIRT5, Tom20, and Tom22, but not SIRT1, were completely abolished by specific siRNAs for both LKB1 and CaMKKβ (Figure 4e). KRGE-induced LKB1 selectively regulated SIRT1, suggesting that Ca^2+^-mediated CaMKKβ activation does not affect SIRT1 expression. By contrast, KRGE-induced LKB1 and CaMKKβ synergistically affected SIRT5, Tom20, and Tom22 expression.

### 3.6. SIRT1 and SIRT5 Synergistically Regulate KRGE-Induced Changes in the Expression of the Mitochondria Outer Membrane Proteins Tom20 and Tom22

To further examine whether KRGE–bilirubin-induced changes in mitochondrial function involve SIRT1–5, we evaluated the expression of SIRT1–5 in the presence of bilirubin or biliverdin. Treatment of astrocytes with bilirubin significantly increased the protein levels of SIRT1 and SIRT5, and these effects were not observed when cells were treated with biliverdin (Figure 5a and Appendix A). We did not find any significantly different levels of SIRT2, SIRT3, and SIRT4 when cells were treated with bilirubin or biliverdin (Figure 5a and Appendix A). The KRGE-induced increases in SIRT1 and SIRT5 protein levels were reduced when the cells were treated with siRNAs against ERRα (Figure 5b). SIRT1 knockdown inhibited the KRGE-induced expression of ERRα, Cyt c, SOD2, Tom20, and Tom22, and these effects were more significant when cells were transfected with both SIRT1 and SIRT5 (Figure 5c). ERRα knockdown also reduced KRGE-induced Tom20 and Tom22 protein levels (Appendix A). These results suggest crosstalk between SIRT1 and ERRα in KRGE-treated astrocytes. We assumed that the LKB1–SIRT1 or CaMKKβ–SIRT5 pathway may not be sufficient to explain the effects of KRGE on mitochondrial function. Instead, SIRT1 and SIRT5 synergistically regulate the expression of mitochondrial functional proteins. By isolating the mitochondrial/cytosolic fraction, we found that SIRT1 was discretely localized in the cytosol, whereas SIRT5, Cyt c, SOD2, Tom20, and Tom22 were discretely localized in mitochondria (Figure 5d). Mitochondria-localized SOD2 upregulation by KRGE was confirmed using immunocytochemistry (Figure 5e, top) without affecting ROS production (Figure 5e, bottom). Thus, KRGE-induced BVR-A/bilirubin may facilitate the interplay between cytosolic SIRT1 and mitochondrial SIRT5 in astrocytes under physiological conditions.

### 3.7. Exogenous Bilirubin Recovers KRGE-Modulated Mitochondrial Function in BVR-A-Depleted Astrocytes

We examined the bilirubin-mediated signaling pathways of mitochondrial function influenced by KRGE. KRGE markedly increased the expression of BVR-A and proteins of its downstream signaling pathways, such as LKB1–SIRT1 and CaMKKβ–SIRT5, leading to Tom20/Tom22-induced increases in Mito, Cyt c, and SOD2 levels. Because BVR-A converts biliverdin to bilirubin under physiological conditions, we next checked whether exogenous bilirubin can recover those mitochondrial signaling molecules when astrocytes are subjected to decreased BVR-A expression. KRGE increased the protein levels of p-LKB1, SIRT1, SIRT5, Tom20, Cyt c, and SOD2; these effects were completely blocked by BVR-A knockdown (Figure 6a). Exogenous bilirubin significantly recovered the levels of these mitochondrial functional proteins (Figure 6a) and OXPHOS (i.e., complex I and complex IV) (Figure 6b) under conditions of reduced BVR-A expression. The localization of Tom20 in mitochondria and its expression pattern assessed by immunocytochemistry (Figure 6c) were similar to the protein levels detected by Western blotting (Figure 6a).

### 3.8. KRGE Induces Mitochondrial Functional Proteins via Tom20/Tom22 in Astrocytes

The combined reduction in Tom20 and Tom22 expression in astrocytes significantly reduced the KRGE-induced Mito, Cyt c, SOD2, and SIRT5 expression (Figure 7a). Under these conditions, SIRT1 expression was not significantly changed by Tom20 or Tom22 (Figure 7a), suggesting that SIRT1 is an upstream signal for Tom20 and Tom22. KRGE upregulated ATP production, which was almost completely blocked by the reduction in Tom20, Tom22, or BVR-A (Figure 7b). The combined reduction of Tom20 and Tom22 did not further reduce KRGE-mediated ATP levels (Figure 7b), suggesting that Tom20 or Tom22 alone is sufficient for KRGE-mediated ATP production. These results suggest that KRGE-regulated Tom20 and Tom22 in mitochondria play key roles in ATP production, mitochondrial mass (i.e., Mito), OXPHOS (i.e., Cyt c), and antioxidative conditions (i.e., SOD2) (Figure 7c).

### 3.9. KRGE Induces Mitochondrial Functional Proteins in Brain Tissues

Similar to in vitro astrocytes, GFAP-positive in vivo astrocytes expressed Tom20 in the fimbria region of the hippocampus (approximately bregma −1.7) when mice were administered KRGE for 3 days (Figure 8a). Brain tissues containing the hippocampus upregulated the ERRα, SIRT1, SIRT5, Tom20, Tom22, Cyt c, and SOD2 protein levels in a KRGE-concentration-dependent manner (Figure 8b), suggesting that KRGE upregulates mitochondrial functional proteins in the brain tissues.

## 4. Discussion

KRGE plays a role in health prophylaxis; however, the molecular mechanisms of active mitochondrial conditions induced by KRGE have not been well established. Our findings suggest that KRGE stimulates astrocytic mitochondrial function partly through bilirubin production. BVR-A can produce bilirubin from biliverdin, which is an HO metabolite. Compared to biliverdin, bilirubin has a more potent effect on mitochondrial membrane potential and mass as well as ERRα activation, consequently enhancing the expression of OXPHOS, Cyt c, and SOD2. SOD2 exists in homotetramers and localizes to the mitochondrial matrix, regulating mitochondrial oxidative stress [22]. KRGE may have antioxidant effects by upregulating the levels of bilirubin, a strong antioxidant, and, consequently, suppressing undesirable oxidative stress through SOD2.

Canto and Auwerx [23] suggest that AMPK, SIRT1, and PGC-1α act as an orchestrated network to improve metabolic fitness and control energy expenditure. Based on this research, we expanded our study to signaling proteins such as ERRα, LKB1, Tom20, Tom22, and SIRT1-5, which are related to mitochondrial functions. The energy metabolism kinase LKB1 helps maintain ATP levels and is critical for neuronal survival following mitochondrial dysfunction [19]. LKB1-deficient astrocytes exhibit impaired metabolic function and enhanced inflammatory activation [18]. Under physiological circumstances, KRGE activates the LKB1–SIRT1–ERRα pathway, thereby stimulating Tom20/Tom22 complex-mediated SIRT5 expression. SIRT5 augments KRGE-induced SIRT1–ERRα signaling. Our results show that the SIRT1/SIRT5 combination is more potent in inducing the expression of mitochondrial proteins such as Cyt c, SOD2, Tom20, and Tom22 than SIRT1 alone. In this study, we found that bilirubin activated LKB1, leading to discrete SIRT1 expression in astrocytes. This signaling pathway was dependent on BVR-A-induced bilirubin generation in KRGE-treated astrocytes.

We previously found that CO/R-induced HO-1 initially increases intracellular Ca^2+^ concentrations by activating L-type voltage-gated Ca^2+^ channels, subsequently leading to CaMKKβ-mediated AMPKα activation [8]. This signaling pathway promotes the expression of hypoxia-inducible factor-1α (HIF-1α), PGC-1α, ERRα, and the vascular endothelial growth factor (VEGF) [8,24]. However, in the present study, KRGE-stimulated BVR-A induction promoted ATP production but did not alter AMPKα phosphorylation. Instead, the BVR-A/bilirubin pathway enhanced LKB1 phosphorylation and consequently increased the levels of SIRT1, Mito, Cyt c, and SOD2. CaMKKβ in KRGE-treated astrocytes appeared to facilitate LKB1-mediated changes in mitochondrial function. The combination of LKB1 and CaMKKβ synergistically regulated KRGE-induced SIRT5, Cyt c, SOD2, Tom20, and Tom22 levels. Because KRGE also stimulates the CaMKKβ–AMPKα pathway, we cannot rule out a possible influence of the Ca^2+^–CaMKKβ–AMPKα–PGC-1α pathway on mitochondrial functions. Additionally, our previous study demonstrated that KRGE improves astrocytic mitochondrial functions such as mitochondrial O_2_ consumption and ATP generation by triggering the HO-1-HIF-1α signaling cascade [1]. In the current study, BVR-A did not alter KRGE-induced HO-1 protein levels. Thus, we speculated that KRGE stimulates multiple signaling pathways involved in mitochondrial biogenesis and activity. Thus, our next study will include KRGE-induced Ca^2+^ signaling and its effects on HIF-1α, VEGF, PGC-1α, and AMPKα.

## 5. Conclusions

In astrocytes under physiological conditions, bilirubin generation may be a key event in antioxidant capacity and mitochondrial activity. Here, we demonstrated more potent effects of bilirubin on mitochondrial functions compared to those of biliverdin. In this step, BVR-A induction is a crucial initiator. Our present data demonstrate that KRGE-induced bilirubin improves astrocytic function by triggering the novel signaling pathway along the LKB1–SIRT1–ERRα axis. The SIRT1–ERRα circuit appears to stimulate mitochondria-localized TOM complexes (i.e., Tom20 and Tom22), SIRT5, OXPHOS, Cyt c, and SOD2. These signaling pathways may facilitate the crosstalk between SIRT5 and Tom20/Tom22, probably by improving the mitochondrial function of astrocytes in the hippocampus of normal adult brains.

## Figures and Tables

**Figure 1 antioxidants-11-01742-f001:**
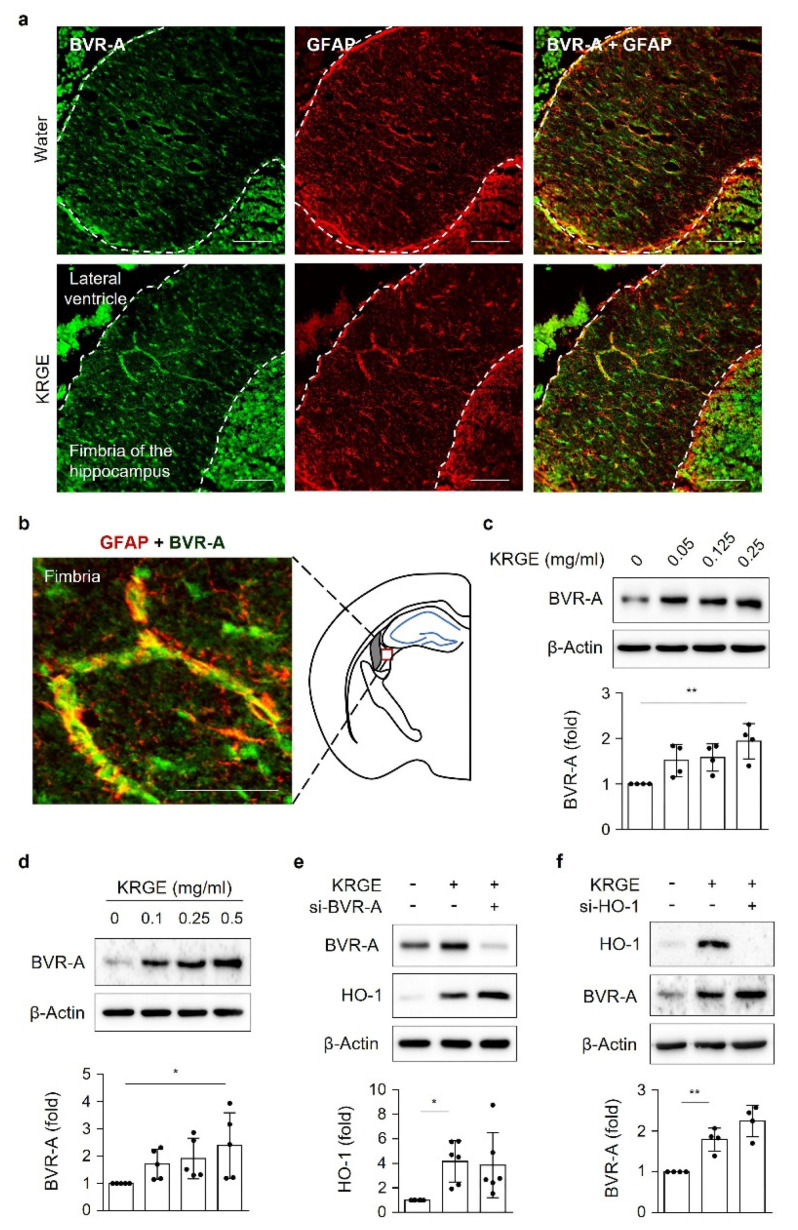
KRGE induces BVR-A and GFAP co-expression in the fimbria region of the mouse hippocampus. (**a**) Representative images of BVR-A (green) and GFAP (red) staining of mouse brains exposed to water or 0.25 mg/mL KRGE for 3 days (*n* = 3 per group). Dotted lines indicate the area of hippocampal fimbria. Scale bar = 100 µm. (**b**) Co-localized staining can be observed as yellow color in the fimbria region of 0.25 mg/mL KRGE-treated mice. Scale bars = 50 µm. (**c**) The expression levels of BVR-A and β-actin proteins in the brain tissues were determined using Western blot analysis, and BVR-A levels were quantified (*n* = 4 per group). (**d**) Astrocytes were incubated with KRGE at various concentrations for 24 h, and BVR-A levels were detected by Western blot analysis (*n* = 5). (**e**,**f**) Astrocytes were transfected with control or the indicated siRNAs (i.e., si-BVR-A or si-HO-1) and subjected to 0.25 mg/mL KRGE for 24 h. HO-1 (*n* = 6) or BVR-A (*n* = 4) protein levels in cell lysates were detected via Western blotting. * *p* < 0.05; ** *p* < 0.01. BVR-A, biliverdin reductase A; GFAP, glial fibrillary acidic protein; HO-1, heme oxygenase 1; KRGE, Korean red ginseng extract; siRNA, small interfering RNA.

**Figure 2 antioxidants-11-01742-f002:**
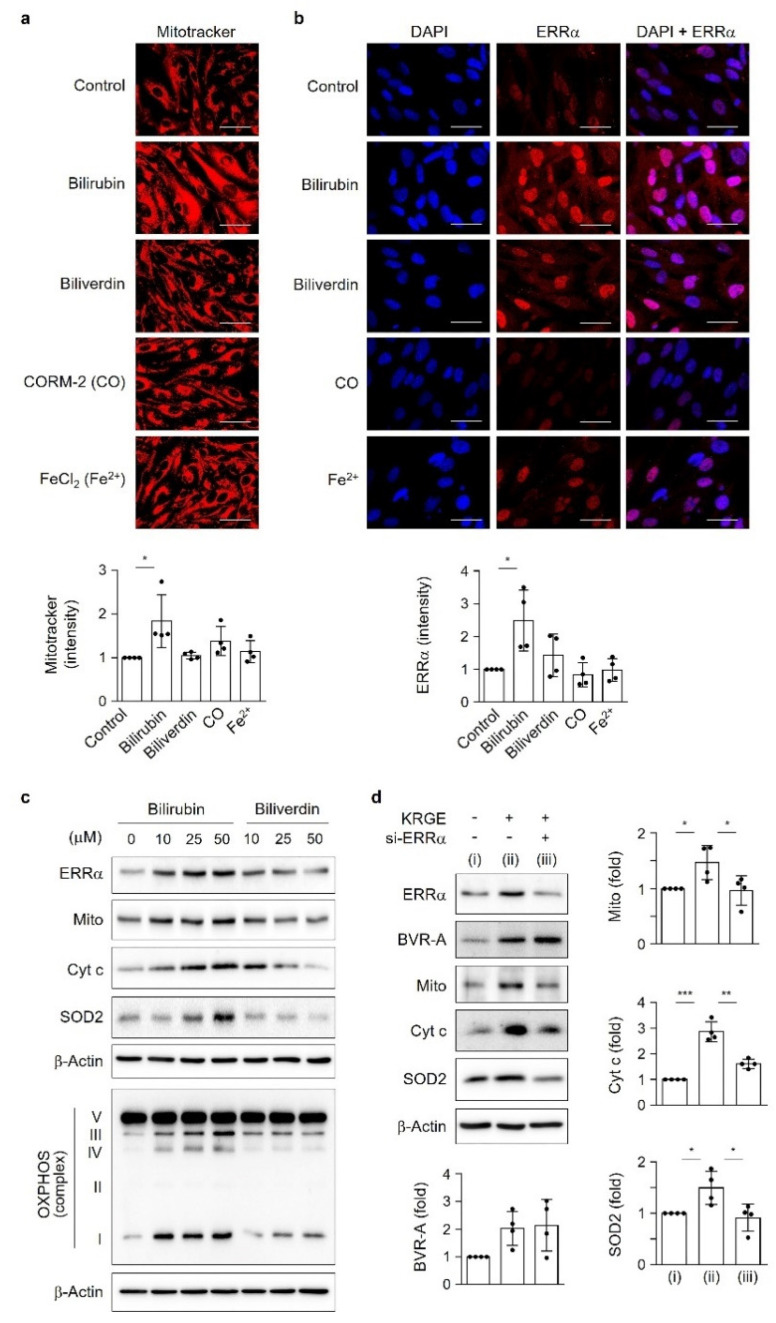
Among HO metabolites, bilirubin significantly induces ERRα expression and mitochondrial function. (**a**) Astrocytes were incubated with 50 µM bilirubin, 50 µM biliverdin, 50 µM CORM-2 (CO), or 50 µM FeCl_2_ (Fe^2+^) for 3.5 h followed by 0.5 µM MitoTracker-Red treatment for 0.5 h. Live cells were visualized using a confocal microscope (*n* = 4). Scale bar = 25 µm. (**b**,**c**) Astrocytes were incubated with 50 µM bilirubin, 50 µM biliverdin, 50 µM CORM-2 (CO), or 50 µM FeCl_2_ (Fe^2+^) for 4 h. (**b**) ERRα expression was detected by immunocytochemistry. Nuclei were stained with DAPI. Scale bar = 25 µm. (**c**) Indicated proteins were detected by Western blotting (*n* = 5–7). (**d**) Astrocytes were transfected with control or indicated siRNAs and subjected to 0.5 mg/mL KRGE for 24 h. Target protein levels in cell lysates were detected via Western blotting (*n* = 4). (i) Water + control siRNA; (ii) KRGE + control siRNA; (iii) KRGE + si-ERRα. * *p* < 0.05; ** *p* < 0.01; *** *p* < 0.001. CO, carbon monoxide; CORM-2, [Ru(CO)_3_Cl_2_]_2_; DAPI, 4′,6-diamidino-2-phenylindole; ERRα, estrogen-related receptor α.

**Figure 3 antioxidants-11-01742-f003:**
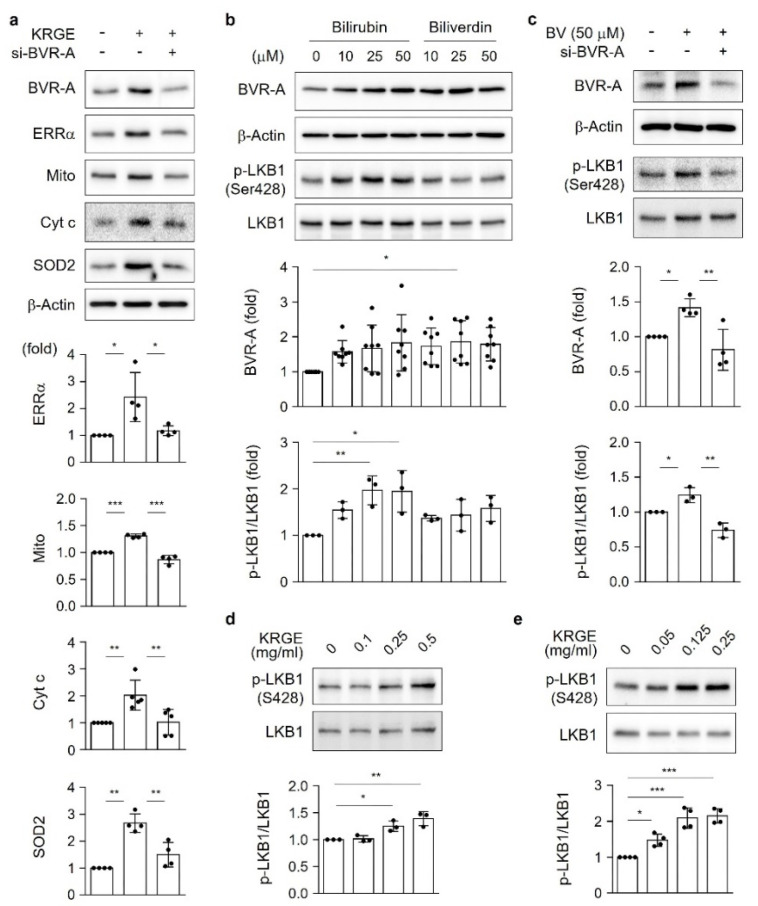
KRGE induces BVR-A/bilirubin-mediated LKB1 activation. (**a**) Astrocytes were transfected with control or the indicated siRNAs and subjected to 0.5 mg/mL KRGE for 24 h. Target protein levels in cell lysates were detected via Western blotting (*n* = 4–5). (**b**) Astrocytes were incubated with 50 µM bilirubin or 50 µM biliverdin for 4 h. BVR-A (*n* = 8), p-LKB (*n* = 3), and LKB1 (*n* = 3) protein levels were assessed by Western blotting. (**c**) Astrocytes were transfected with control or the indicated siRNAs and subjected to KRGE treatment. Target protein levels in whole-cell lysates were detected via Western blotting (*n* = 3–4). (**d**) Cells were treated with various concentrations of KRGE for 24 h. p-LKB1 and LKB1 protein levels were detected (*n* = 3). (**e**) The expression of p-LKB1 and LKB1 proteins was determined in brain tissues using Western blot analysis, and p-LKB1/LKB1 ratios were quantified (*n* = 4 per group). * *p* < 0.05; ** *p* < 0.01; *** *p* < 0.001. LKB1, liver kinase B1.

**Figure 4 antioxidants-11-01742-f004:**
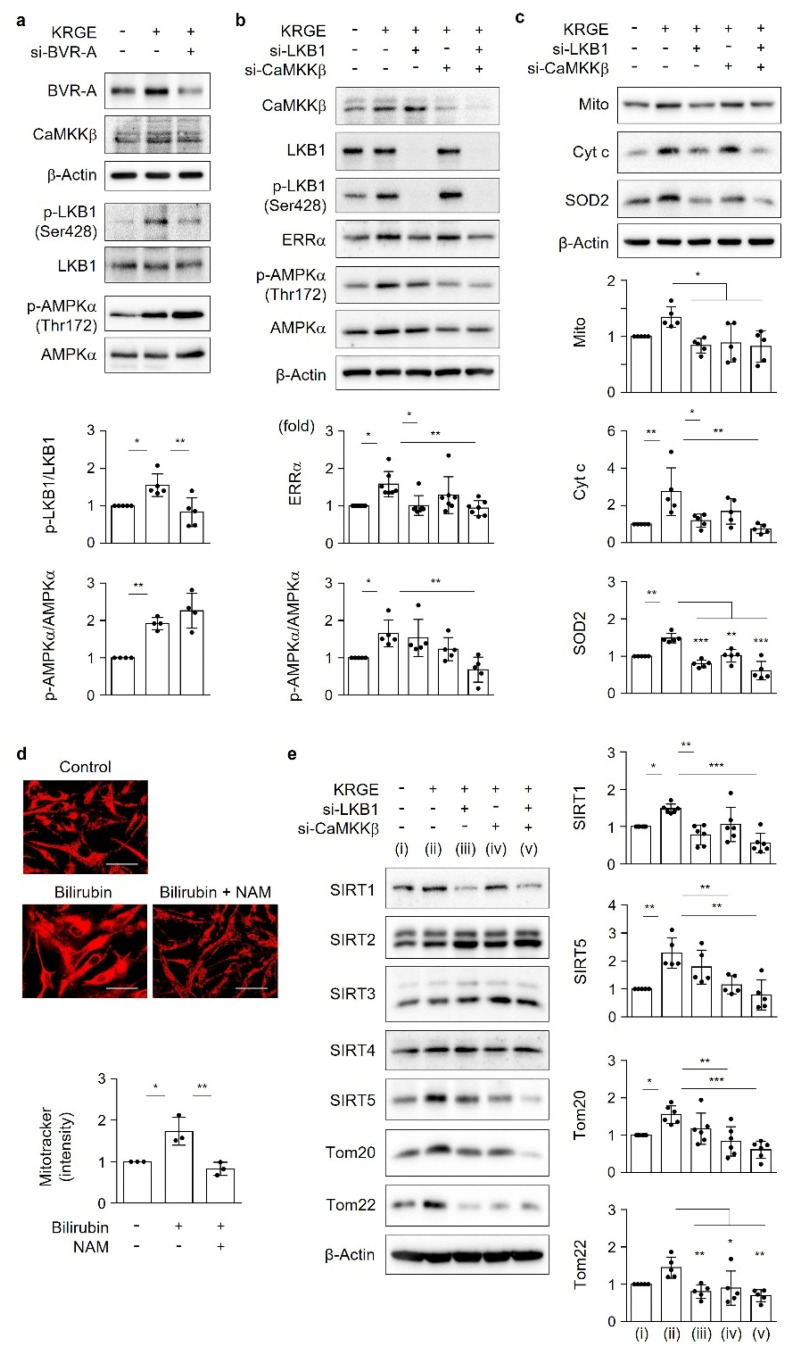
KRGE induces the LKB1-mediated expression of ERRα and mitochondrial functional proteins. (**a**–**c**) Astrocytes were transfected with control or target siRNAs and subjected to KRGE. Indicated protein levels were analyzed using Western blotting and quantified (*n* = 4–6). (**d**) Astrocytes were incubated with 5 mM NAM for 30 min, followed by 50 µM bilirubin for 3.5 h and 0.5 µM MitoTracker-Red for 0.5 h. Live astrocyte cells were imaged using a confocal microscope (*n* = 3). Scale bar = 25 µm. (**e**) Astrocytes were transfected with control or the indicated siRNAs, followed by 0.5 mg/mL KRGE treatment for 24 h. Target proteins were analyzed by Western blotting (*n* = 5–6). (i) Water + control siRNA; (ii) KRGE + control siRNA; (iii) KRGE + si-LKB1; (iv) KRGE + si-CaMKKβ; (v) KRGE + si-LKB1 + si-CaMKKβ. * *p* < 0.05; ** *p* < 0.01; *** *p* < 0.001. CaMKKβ, Ca^2+^/calmodulin-dependent protein kinase kinase β; NAM, nicotinamide.

**Figure 5 antioxidants-11-01742-f005:**
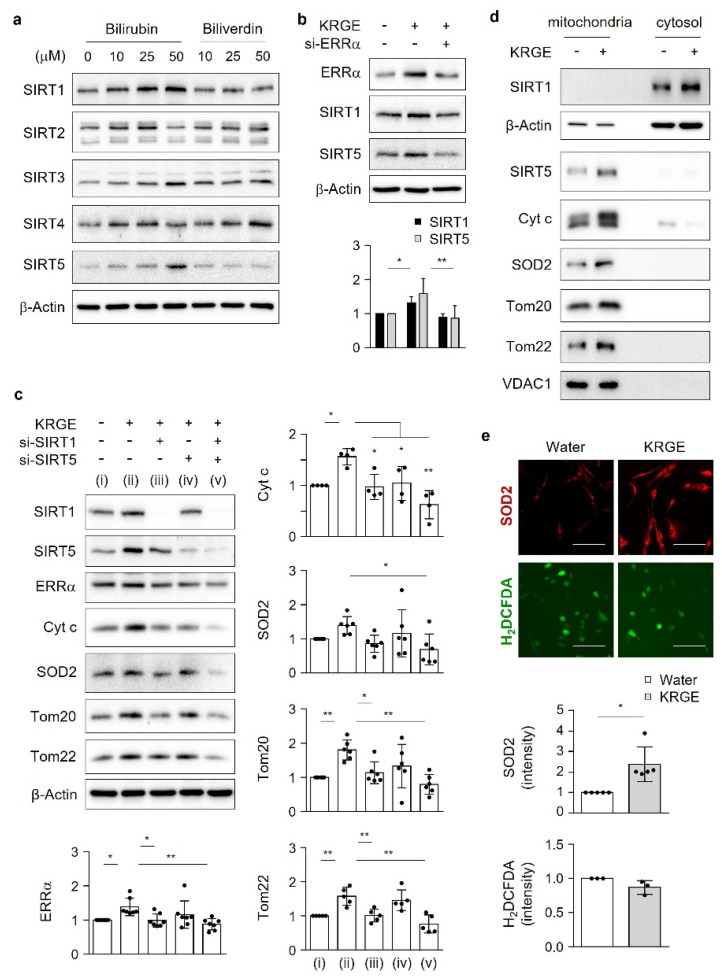
SIRT1 and SIRT5 synergistically regulate the KRGE-induced expression of the mitochondria outer membrane proteins Tom20 and Tom22. (**a**) Astrocytes were incubated with 50 µM bilirubin or 50 µM biliverdin for 4 h. Indicated protein levels were assessed by Western blotting (*n* = 5–6). (**b**,**c**) Astrocytes were transfected with control or target siRNAs, followed by administration of 0.5 mg/mL KRGE for 24 h. (**b**) Indicated protein levels (i.e., SIRT1 (*n* = 5) and SIRT5 (*n* = 6)) were detected by Western blotting and quantified. (**c**) (i) Water + control siRNA; (ii) KRGE + control siRNA; (iii) KRGE + SIRT1 siRNA (si-SIRT1); (iv) KRGE + si-SIRT5; (v) KRGE + si-SIRT1 + si-SIRT5. Indicated protein levels were detected by Western blotting and quantified (*n* = 4–6 independent experiments). (**d**) Indicated proteins were obtained from cellular fractions separated into cytosol and mitochondria and detected by Western blotting (*n* = 7). (**e**) Astrocytes were treated with water or 0.5 mg/mL KRGE for 24 h. SOD2 expression was detected by immunocytochemistry (*n* = 5). H_2_DCFDA fluorescence demonstrating cellular ROS was determined (*n* = 3). Scale bar = 100 µm. * *p* < 0.05; ** *p* < 0.01. ROS, reactive oxygen species; SIRT, sirtuin; SOD2, superoxide dismutase 2; Tom, translocase of the outer mitochondrial membrane.

**Figure 6 antioxidants-11-01742-f006:**
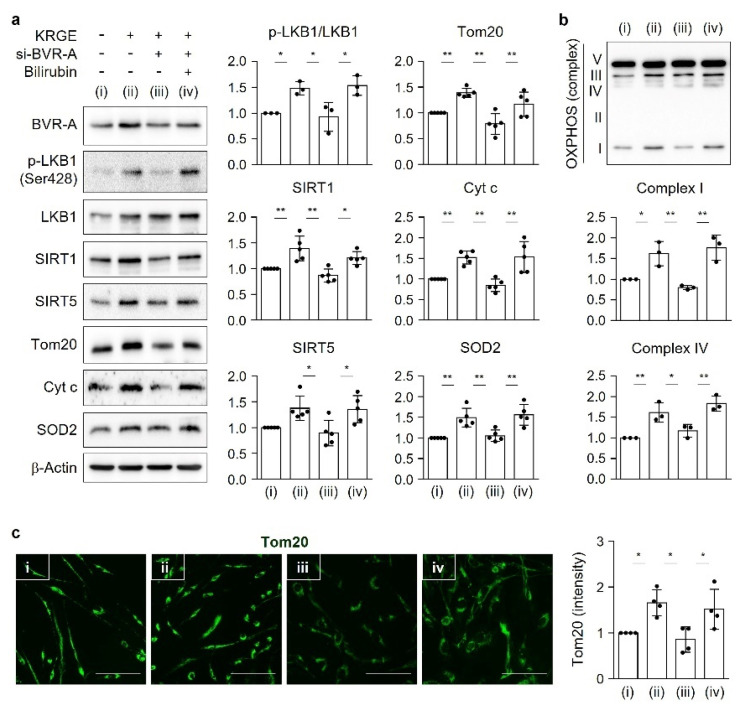
Exogenous bilirubin recovers KRGE-modulated mitochondrial function in BVR-A-depleted astrocyte cells. (**a**–**c**) Astrocytes were transfected with control or BVR-A siRNA. Then, KRGE (0.5 mg/mL) was added, and the cells were incubated for 20 h, followed by addition of 50 µM bilirubin for 4 h. (i) Water + control siRNA; (ii) KRGE + control siRNA; (iii) KRGE + si-BVR-A; (iv) KRGE + si-BVR-A + bilirubin. (**a**,**b**) Indicated protein levels were detected by Western blotting and quantified (*n* = 5–6). (**c**) Tom20 expression was detected by immunocytochemistry (*n* = 4). Scale bar = 100 µm. * *p* < 0.05; ** *p* < 0.01.

**Figure 7 antioxidants-11-01742-f007:**
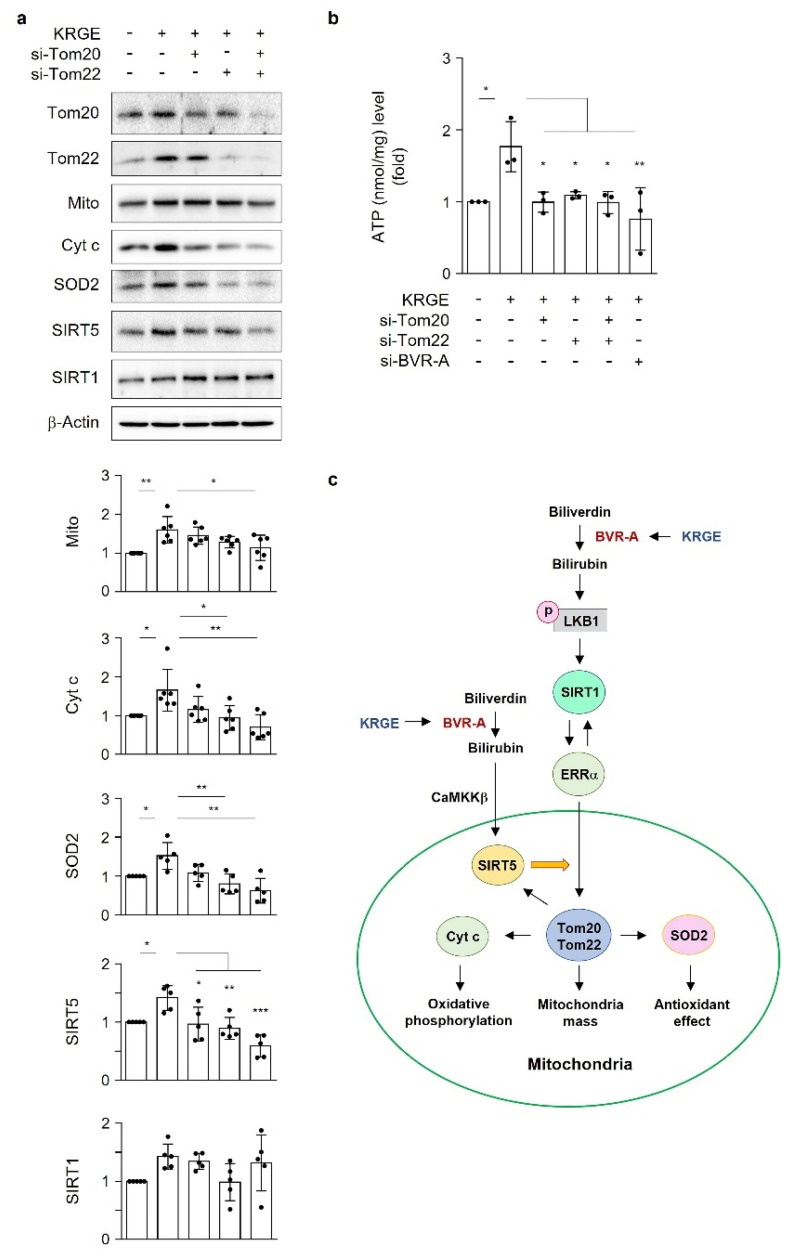
KRGE induces the expression of mitochondrial functional proteins in astrocytes via Tom20/Tom22. (**a**) Astrocytes were transfected with control or the indicated siRNAs. Target protein levels were detected via Western blotting, and the protein levels were quantified (*n* = 5–6). (**b**) Astrocytes were transfected with control or the indicated siRNAs. ATP levels (nmol/mg) were detected (*n* = 3). (**c**) Schematic figure showing the molecular mechanisms of how KRGE regulates mitochondrial functions in astrocytes under physiological conditions. * *p* < 0.05; ** *p* < 0.01; *** *p* < 0.001.

**Figure 8 antioxidants-11-01742-f008:**
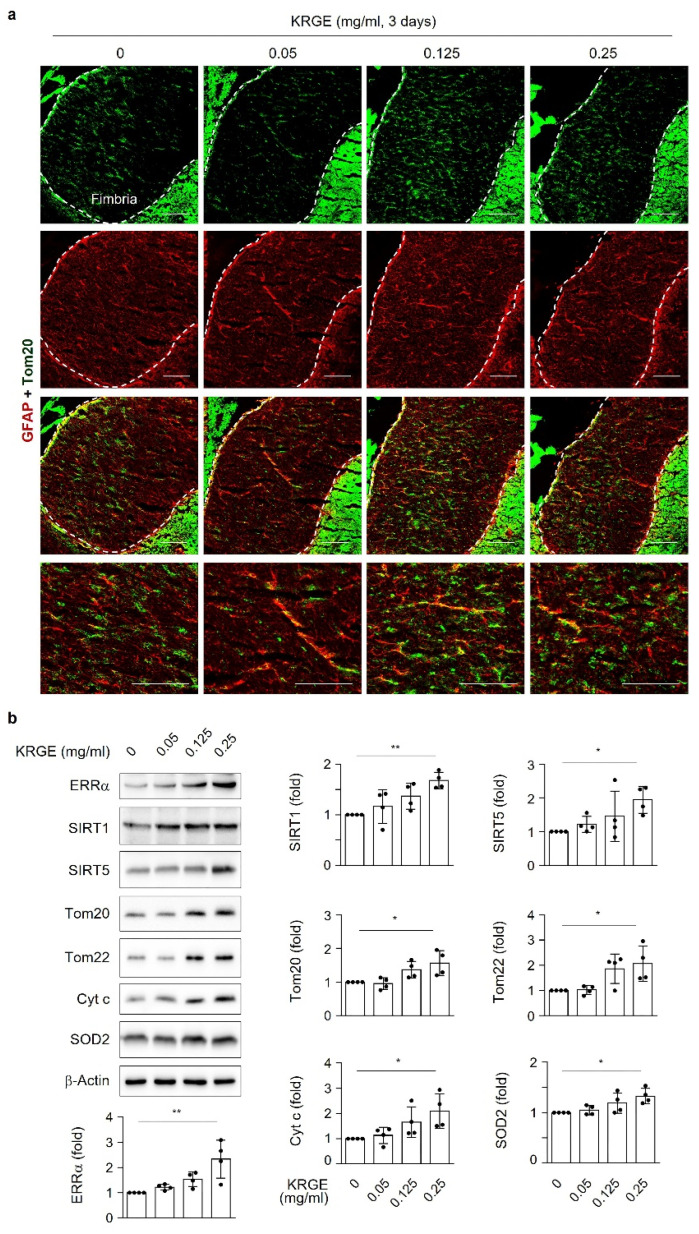
KRGE administration upregulated mitochondrial functional proteins. (**a**) Representative images of Tom20- (green) and GFAP-stained (red) mouse brains following water or KRGE administration for 3 days (*n* = 3 per group). Dotted lines indicate the fimbria region of the hippocampus. Co-localized staining can be observed as yellow color in the fimbria region. Scale bars = 100 µm. (**b**) The expression levels of target proteins were determined in brain tissues using Western blot analysis (*n* = 4 per group). * *p* < 0.05; ** *p* < 0.01.

## Data Availability

The data presented in this study are contained within the article. Original data will be made available on request from the corresponding author.

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
