# Peer review of "Induction of BVR-A Expression by Korean Red Ginseng in Murine Hippocampal Astrocytes: Role of Bilirubin in Mitochondrial Function via the LKB1–SIRT1–ERRα Axis"

_antioxidants, 2022, doi:10.3390/antiox11091742_

Round 1

Reviewer 1 Report

Interesting study.

Reviewer 2 Report

In this manuscript, the authors show that Korean red ginseng extract (KRGE) induces biliverdin reductase A (BVR-A) in astrocytes both in vitro and in vivo. They then explore the role of BVR-A induction and its downstream pathways in some of the effects of KRGE on mitochondria. Although the results are interesting, there are a number of points that need to be addressed. First, while the in vitro studies use a variety of siRNAs and include multiple readouts, the authors never actually look at mitochondrial function although throughout they say that KGRE improves mitochondrial function. This is a major problem as there is no evidence provided that KRGE does improve mitochondrial function as shown, at a minimum, by effects on mitochondrial bioenergetics and ATP production or that BVR-A and its downstream pathways play a role in this improvement. Second, the authors do not explain well why they chose to look at specific signaling pathways. For example, the authors never indicate why they chose initially to look at ERRa and its induction by KRGE. Nor do they explain why they initially thought that SIRTs could be downstream of LKB and CaMKKb. They also don’t explain why they only look at BVR-A, phosphoLKB, SIRT1, SIRT5, Tom20 and Tom22 in the brain samples. They should look at the entire pathway that they propose in these samples and, ideally, do it in a single figure rather than scattered throughout the manuscript. Other points that need to be addressed are listed below.

1. Methods, line 191: What protein or proteins does the mitochondria antibody recognize?

2. Figure 1a: The authors need to outline the area of the fimbria in each of the pictures.

3. Experiments with bilirubin and biliverdin. The concentration of 50 µM seems very high. How does this correlate with the physiological levels of both molecules in the brain? The authors need to justify the use of this concentration.

4. lines 337-338: In Fig. 4e there is no indication that KGRE significantly increases SIRT1 and the effect of KGRE on SIRT1 seems to be quite variable throughout. The authors need to address this point.

5. Figure 6a needs to include GFAP staining alone and the different rows should be clearly labeled. Also, as in Figure 1a, the fimbria area should be outlined in each of the pictures.

6. lines 404-405: The authors need to make clear that they are switching from brain samples to fibroblasts here. This transition is very abrupt and confusing to the reader.

7. Figure 7a seems to repeat many of the experiments shown earlier. The only difference is the inclusion of bilirubin. Why didn’t the authors just include bilirubin in the earlier experiments which would not only make a lot more sense but also make the manuscript flow better.

Author Response

Reviewer #1

In this manuscript, the authors show that Korean red ginseng extract (KRGE) induces biliverdin reductase A (BVR-A) in astrocytes both in vitro and in vivo. They then explore the role of BVR-A induction and its downstream pathways in some of the effects of KRGE on mitochondria. Although the results are interesting, there are a number of points that need to be addressed.

Comment #1: First, while the in vitro studies use a variety of siRNAs and include multiple readouts, the authors never actually look at mitochondrial function although throughout they say that KGRE improves mitochondrial function. This is a major problem as there is no evidence provided that KRGE does improve mitochondrial function as shown, at a minimum, by effects on mitochondrial bioenergetics and ATP production or that BVR-A and its downstream pathways play a role in this improvement.

Response #1: Per your valuable suggestion, we performed experiments for ATP detection. KRGE-mediated enhanced ATP levels were blocked by BVR-A siRNA (si-BVR-A) as well as si-Tom20 and/or si-Tom22 (Figure 7b).

Comment #2: Second, the authors do not explain well why they chose to look at specific signaling pathways. For example, the authors never indicate why they chose initially to look at ERRa and its induction by KRGE. Nor do they explain why they initially thought that SIRTs could be downstream of LKB and CaMKKb.

Response #2: We added explanation in Discussion Section (Line 490-493).

Canto and Auwerx [23] suggest that AMPK, SIRT1, and PGC-1α acts as an orchestrated network to improve metabolic fitness and to control energy expenditure. Based on this research, we expanded signaling proteins such as ERRa, LKB1, Tom20, Tom22, and SIRT1-5, which are related to mitochondrial functions.

[23]   Canto, C.; Auwerx, J., PGC-1alpha, SIRT1 and AMPK, an energy sensing network that controls energy expenditure. Curr Opin Lipidol 2009, 20, (2), 98-105.

The relationship among ERRa, PGC-1a, LKB1, CaMKKb, AMPKa, SIRT1-5, Tom20, Tom22 and HIF-1a is dynamic, complex, and sophisticated. We are trying to find out the exact mechanism how KRGE regulates those mitochondria-related proteins in pathophysiologic conditions. So far, our novel finding shows that LKB1 but not CaMKKb can be an upstream factor for KRGE-induced SIRT1 during physiologic conditions. Further studies may need to be necessary to figure out the exact mechanism.

Comment #3: They also don’t explain why they only look at BVR-A, phosphor LKB, SIRT1, SIRT5, Tom20 and Tom22 in the brain samples. They should look at the entire pathway that they propose in these samples and, ideally, do it in a single figure rather than scattered throughout the manuscript.

Response #3: We tried to combine them in Figure 8.

Other points that need to be addressed are listed below.

Comment #4: Methods, line 191: What protein or proteins does the mitochondria antibody recognize?

Response #4: We used the mitochondria antibody (ab92824) purchased from Abcam. According to the manufacturer's description (https://www.abcam.com/mitochondria-antibody-113-1-bsa-and-azide-free-ab92824.html), this antibody is designed to detect human mitochondrial extract. They do not explain which epitope it reacts with on human mitochondria. Instead, it is specifically related to multiple proteins, not just one protein of mitochondria.

Comment #5: Figure 1a: The authors need to outline the area of the fimbria in each of the pictures.

Response #5: We outlined the area of the fimbria in each of the pictures shown in Figure 1a and Figure 8a.

Comment #6: Experiments with bilirubin and biliverdin. The concentration of 50 μM seems very high. How does this correlate with the physiological levels of both molecules in the brain? The authors need to justify the use of this concentration.

Response #6: Moderate concentrations (10 ~ 50 mM) of biliverdin and bilirubin have beneficial effects in various cells [1, 2]. 50 μM Bilirubin (SigmaAldrich, B-4126, m.w. 584.66) concentration in our study is equal to 2.92 mg/dL. Recent report shows the beneficial effects of exercise on plasma bilirubin levels using high capacity running rat model (plasma bilirubin concentration is 2.5-3 mg/dL) [3]. Bilirubin acts as a toxin at high concentrations (higher than approximately 300 mM), especially in neonates, leading to mitochondrial dysfunction and cell death [4, 5].

  1. Stec, D. E.; John, K.; Trabbic, C. J.; Luniwal, A.; Hankins, M. W.; Baum, J.; Hinds, T. D., Jr., Bilirubin Binding to PPARalpha Inhibits Lipid Accumulation. PLoS One 2016, 11, (4), e0153427.
  2. Mancuso, C.; Bonsignore, A.; Di Stasio, E.; Mordente, A.; Motterlini, R., Bilirubin and S-nitrosothiols interaction: evidence for a possible role of bilirubin as a scavenger of nitric oxide. Biochemical pharmacology 2003, 66, (12), 2355-63.
  3. Hinds, T. D., Jr.; Creeden, J. F.; Gordon, D. M.; Spegele, A. C.; Britton, S. L.; Koch, L. G.; Stec, D. E., Rats Genetically Selected for High Aerobic Exercise Capacity Have Elevated Plasma Bilirubin by Upregulation of Hepatic Biliverdin Reductase-A (BVRA) and Suppression of UGT1A1. Antioxidants (Basel) 2020, 9, (9).
  4. Hansen, T. W. R.; Wong, R. J.; Stevenson, D. K., Molecular Physiology and Pathophysiology of Bilirubin Handling by the Blood, Liver, Intestine, and Brain in the Newborn. Physiol Rev 2020, 100, (3), 1291-1346.
  5. Ernster, L.; Zetterstrom, R., Bilirubin, an uncoupler of oxidative phosphorylation in isolated mitochondria. Nature 1956, 178, (4546), 1335-7.

Comment #7: lines 337-338: In Fig. 4e there is no indication that KGRE significantly increases SIRT1 and the effect of KGRE on SIRT1 seems to be quite variable throughout. The authors need to address this point.

Response #7: We did more experiments and added data in Figure 4e.

Comment #8: Figure 6a needs to include GFAP staining alone and the different rows should be clearly labeled. Also, as in Figure 1a, the fimbria area should be outlined in each of the pictures.

Response #8: We added GFAP alone data in Figure 8a and outlined the fimbria area in Figure 1a and Figure 8a.

Comment #9: lines 404-405: The authors need to make clear that they are switching from brain samples to fibroblasts here. This transition is very abrupt and confusing to the reader.

Comment #9: We rearranged Figure 7 and Figure 8. Figure 7 demonstrates in vitro cell data while Figure 8 shows in vivo brain tissues’ data.

Comment #10: Figure 7a seems to repeat many of the experiments shown earlier. The only difference is the inclusion of bilirubin. Why didn’t the authors just include bilirubin in the earlier experiments which would not only make a lot more sense but also make the manuscript flow better.

Comment #10: We moved Figure 7 into Figure 6 per your valuable comment. 

Reviewer 3 Report

The manuscript by moon et al., shows an important role for LKB1-SIRT1-ERRα axis as an underlying molecular mechanism in Korean red ginseng extract (KRGE) induced mitochondrial function in astrocytes and mouse hippocampus. The study uncovers a novel mechanism via which KRGE exerts its effects on mitochondrial function in astrocytes. Please see my minor comments noted below and the revised manuscript can be accepted for a publication in Antioxidants.

Minor comments

1.      Line 232: Please correct the typo: BVA-A to BVR-A

2.      Figures 2d &7a,b&c: Since the main figure panels are labeled with letters; a,b,c….I recommend changing the group labels in figure 2d &7a,b&c to a different form of labeling (i.e., I, II, II) to avoid confusion.

3.      Figure 5f: authors used GAPDH as a loading control for a western blot in isolated mitochondria. GAPDH is not expressed in mitochondria. Mitochondrial proteins such VDAC are commonly used as loading control for mitochondrial isolates.

4.      Measuring functional outcomes such as mitochondrial respiration and ROS production in this experimental set up would have been a great addition to the existing the data presented in the manuscript, although not necessary.

Author Response

Reviewer #2

The manuscript by moon et al., shows an important role for LKB1-SIRT1-ERRα axis as an underlying molecular mechanism in Korean red ginseng extract (KRGE) induced mitochondrial function in astrocytes and mouse hippocampus. The study uncovers a novel mechanism via which KRGE exerts its effects on mitochondrial function in astrocytes. Please see my minor comments noted below and the revised manuscript can be accepted for a publication in Antioxidants.

Minor comments

  1. Line 232: Please correct the typo: BVA-A to BVR-A

Response #1: We changed it.

  1. Figures 2d &7a,b&c: Since the main figure panels are labeled with letters; a,b,c….I recommend changing the group labels in figure 2d &7a,b&c to a different form of labeling (i.e., I, II, II) to avoid confusion.

Response #2: We have changed them per your suggestion (i.e., (i), (ii), (iii) etc.).

  1. Figure 5f: authors used GAPDH as a loading control for a western blot in isolated mitochondria. GAPDH is not expressed in mitochondria. Mitochondrial proteins such VDAC are commonly used as loading control for mitochondrial isolates.

Response #3: To make clear data, we changed GAPDH data into b-Actin and VDAC1.

  1. Measuring functional outcomes such as mitochondrial respiration and ROS production in this experimental set up would have been a great addition to the existing the data presented in the manuscript, although not necessary.

Response #4: We added ROS data (Figure 5e) and detected ATP levels (Figure 7b) per your valuable comment.

Round 2

Reviewer 2 Report

The authors have addressed all of my concerns.